# Behavioral training promotes multiple adaptive processes following acute hearing loss

Peter Keating*[†], Onayomi Rosenior-Patten, Johannes C Dahmen, Olivia Bell, Andrew J King

Department of Physiology, Anatomy and Genetics, University of Oxford, Oxford, United Kingdom

**Abstract** The brain possesses a remarkable capacity to compensate for changes in inputs resulting from a range of sensory impairments. Developmental studies of sound localization have shown that adaptation to asymmetric hearing loss can be achieved either by reinterpreting altered spatial cues or by relying more on those cues that remain intact. Adaptation to monaural deprivation in adulthood is also possible, but appears to lack such flexibility. Here we show, however, that appropriate behavioral training enables monaurally-deprived adult humans to exploit both of these adaptive processes. Moreover, cortical recordings in ferrets reared with asymmetric hearing loss suggest that these forms of plasticity have distinct neural substrates. An ability to adapt to asymmetric hearing loss using multiple adaptive processes is therefore shared by different species and may persist throughout the lifespan. This highlights the fundamental flexibility of neural systems, and may also point toward novel therapeutic strategies for treating sensory disorders.

**\*For correspondence:** peter. keating@dpag.ox.ac.uk

**Present address:** [†]Ear Institute, University College London, London, United Kingdom

## Introduction

A major challenge faced by the brain is to maintain stable and accurate representations of the world despite changes in sensory input. This is important because the statistical structure of sensory experience varies across different environments (*Mlynarski and Jost, 2014*; *Qian et al., 2012*; *Seydell et al., 2010*), but also because long-term changes in sensory input result from a range of sensory impairments (*Feldman and Brecht, 2005*; *Keating and King, 2015*; *Sengpiel, 2014*). Adaptation to altered inputs has been demonstrated in different sensory systems, particularly during development, and serves to shape neural circuits to the specific inputs experienced by the individual (*Margolis et al., 2014*; *Mendonca, 2014*; *Schreiner and Polley, 2014*; *Sur et al., 2013*). However, many ecologically important aspects of neural processing require the integration of multiple sensory cues, either within or across different sensory modalities (*Seilheimer et al., 2014*; *Seydell et al., 2010*). A specific change in sensory input may therefore have a considerable impact on some cues whilst leaving others intact. In such cases, adaptation can be achieved in two distinct ways, as demonstrated by recent studies of sound localization following monaural deprivation during infancy (*Keating et al., 2013*; *2015*).

Monaural deprivation alters the binaural spatial cues that normally determine the perceived location of a sound in the horizontal plane (*Figure 1A*) (*Kumpik et al., 2010*; *Lupo et al., 2011*). Adaptation can therefore be achieved by learning the altered relationships between particular cue values and spatial locations (*Gold and Knudsen, 2000*; *Keating et al., 2015*; *Knudsen et al., 1984*), a process referred to as cue remapping. However, at least in mammals, monaural spectral cues are also available to judge sound source location in both the horizontal and vertical planes (*Carlile et al., 2005*). These spectral cues arise from the acoustic properties of the head and external ears, which

**eLife digest** The brain normally compares the timing and intensity of the sounds that reach each ear to work out a sound's origin. Hearing loss in one ear disrupts these between-ear comparisons, which causes listeners to make errors in this process. With time, however, the brain adapts to this hearing loss and once again learns to localize sounds accurately.

Previous research has shown that young ferrets can adapt to hearing loss in one ear in two distinct ways. The ferrets either learn to remap the altered between-ear comparisons, caused by losing hearing in one ear, onto their new locations. Alternatively, the ferrets learn to locate sounds using only their good ear. Each strategy is suited to localizing different types of sound, but it was not known how this adaptive flexibility unfolds over time, whether it persists throughout the lifespan, or whether it is shared by other species.

Now, Keating et al. show that, with some coaching, adult humans also adapt to temporary loss of hearing in one ear using the same two strategies. In the experiments, adult humans were trained to localize different kinds of sounds while wearing an earplug in one ear. These sounds were presented from 12 loudspeakers arranged in a horizontal circle around the person being tested. The experiments showed that short periods of behavioral training enable adult humans to adapt to a hearing loss in one ear and recover their ability to localize sounds.

Just like the ferrets, adult humans learned to correctly associate altered between-ear comparisons with their new locations and to rely more on the cues from the unplugged ear to locate sound. Which of these adaptive strategies the participants used depended on the frequencies present in the sounds. The cells in the ear and brain that detect and make sense of sound typically respond best to a limited range of frequencies, and so this suggests that each strategy relies on a distinct set of cells. Keating et al. confirmed in ferrets that different brain cells are indeed used to bring about adaptation to hearing loss in one ear using each strategy. These insights may aid the development of new therapies to treat hearing loss.

filter sounds in a direction-dependent way (*Figure 1B*). Monaural deprivation has no effect on the spectral cues available to the non-deprived ear. This means it is possible to adapt by learning to rely more on these unchanged spectral cues, whilst learning to ignore the altered binaural cues (*Kacelnik et al., 2006*; *Keating et al., 2013*; *Kumpik et al., 2010*), a form of adaptation referred to as cue reweighting.

Developmental studies of sound localization plasticity following monaural deprivation have found evidence for both cue remapping (*Gold and Knudsen, 2000*; *Keating et al., 2015*; *Knudsen et al., 1984*) and cue reweighting (*Keating et al., 2013*), but it is not known whether these adaptive processes can occur simultaneously. Indeed, until recently, it was thought that monaural deprivation might induce different adaptive processes in different species (*Keating and King, 2013*; *Shamma, 2015*). However, whilst we now know that ferrets use both cue remapping and reweighting to adapt to monaural deprivation experienced during development (*Keating et al., 2013*; *2015*), it is not known whether the same neural populations are involved in each case.

It is also not known whether the ability to use both adaptive processes is restricted to specific species or developmental epochs. Although the mature auditory system can adapt to monaural deprivation using cue reweighting (*Kumpik et al., 2010*), conflicting evidence for cue remapping has been obtained in adult humans fitted with an earplug in one ear for several days (*Florentine, 1976*; *McPartland et al., 1997*). To the extent that adaptive changes in binaural cue sensitivity are possible in adulthood, as suggested by other sensory manipulations (*Trapeau and Schonwiesner, 2015*), these may occur at the expense of cue reweighting. It is therefore unclear whether the same adult individuals can adapt to a unilateral hearing loss using multiple adaptive processes. Although numerous studies have shown that spatial hearing is more plastic early in life (*Keating and King, 2013*; *Knudsen et al., 1984*; *Popescu and Polley, 2010*), behavioral training can facilitate accommodation to altered cues in adulthood (*Carlile, 2014*; *Carlile et al., 2014*; *Kacelnik et al., 2006*; *Shinn-Cunningham et al., 1998*). Here, we show that adult humans are equally capable of using both adaptive processes, provided they are given appropriate training. Moreover, our results suggest

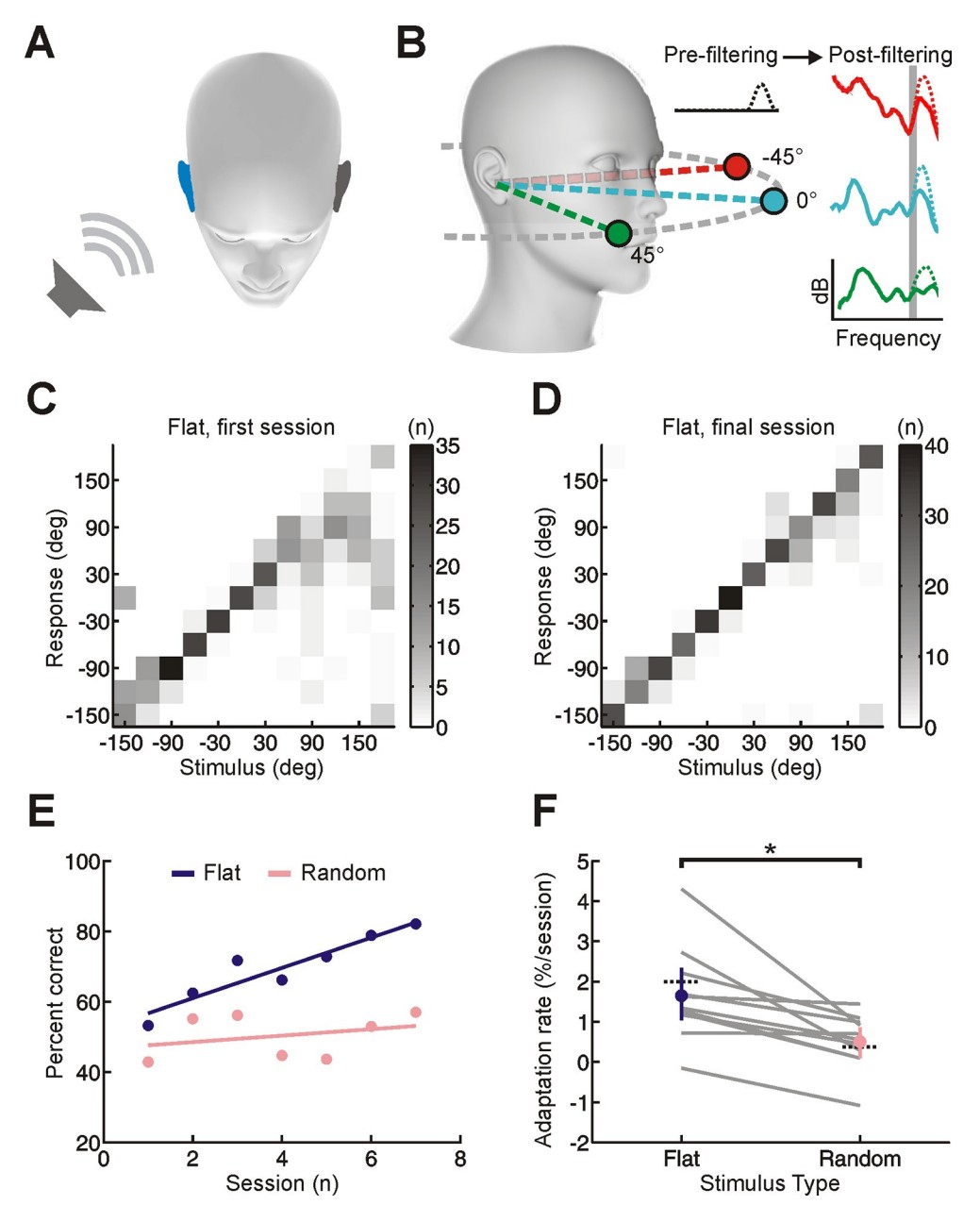

**Figure 1.** Effect of training on localization of broadband noise stimuli in the horizontal plane by monaurally deprived human listeners. (A) When one of these sounds is presented on one side of the head, it will be louder and arrive earlier at the ipsilateral ear (blue), producing interaural time and level differences, which are respectively the primary cues to sound location at low and high frequencies. (B) Because of acoustic filtering by the head and ears, the spectrum of a sound at the tympanic membrane (post-filtering, color) differs from that of the original sound (pre-filtering, black) and varies with location (amplitude in dB is plotted as a function of frequency; color indicates different locations). These spectral cues make it possible to localize sounds using a single ear, but only for sounds that have relatively flat spectra (solid lines) and are sufficiently broadband (shape of spectra in narrow frequency bands varies little with location – see shaded gray region). When spectral features are artificially added to the pre-filtered sound source (dotted lines), these added features can be misattributed to the filtering effects of the head and ears. This produces sound localization errors (e.g. dotted green spectrum is more easily confused with solid turquoise spectrum because of additional peak at high frequencies). The extent of these errors allows us to infer subjects' reliance on spectral cues. (C, D) Joint distribution of stimulus and response obtained from the first (C) and last (D) training session for an individual subject with an earplug in the right ear. Grayscale indicates

*Figure 1 continued on next page*

*Figure 1 continued*

the number of trials corresponding to each stimulus-response combination. Data are shown for trials on which flat-spectrum stimuli were used (i.e. all spatial cues were available). (E) Sound localization performance (% correct) as a function of training session for the same subject. Scores for each session (dots) were fitted using linear regression (lines) to calculate slope values, which quantified the rate of adaptation. Relative to flat-spectrum stimuli (blue), much less adaptation occurred with random-spectrum stimuli (pink), which limit the usefulness of spectral cues to sound location (*Figure 1—figure supplement 1*). (F) Adaptation rate is shown for flat- and random-spectrum stimuli for each subject (gray lines; n = 11). Positive values indicate improvements in localization performance with training. Mean adaptation rates across subjects (± bootstrapped 95% confidence intervals) are shown in blue and pink. Similar results are observed if front-back errors are excluded and changes in error magnitude are calculated (*Figure 1—figure supplement 2*). Dotted black lines indicate adaptation rates observed previously in humans (*Kumpik et al., 2010*; total adaptation reported divided by number of sessions, n = 8).

The following figure supplements are available for figure 1:

**Figure supplement 1.** Experimental setup and stimuli.

**Figure supplement 2.** Effect of training on localization by human listeners of broadband stimuli using same analysis method as for narrowband stimuli in *Figure 2*.

that cue remapping and reweighting are neurophysiologically distinct, which we confirmed by recording from auditory cortical neurons in ferrets reared with an intermittent hearing loss in one ear.

## Results

Adult humans were trained to localize sounds from 12 loudspeakers in the horizontal plane (*Figure 1—figure supplement 1A*) whilst wearing an earplug in one ear (~5600 trials split into 7 sessions completed in < 3 weeks). In order to directly measure the efficacy of training, earplugs were worn only during training sessions. This contrasts with previous work in which adult humans received minimal training, but were required to wear earplugs for extended periods of everyday life (*Florentine, 1976*; *Kumpik et al., 2010*; *McPartland et al., 1997*). On ~50% of trials, subjects were required to localize flat-spectrum broadband noise (0.5–20 kHz), which provide all of the available auditory spatial cues (*Blauert, 1997*). With these cue-rich stimuli, trials were repeated following incorrect responses ("correction trials") and subjects were given performance feedback. Across training sessions, sound localization performance (% correct) gradually improved (*Figure 1C–F*; slope values >0; bootstrap test, p<0.01; Cohen's d = 1.43), indicating that relatively short periods of training are sufficient to drive adaptation.

To determine the relative contributions of cue remapping and reweighting to these changes in localization accuracy, we measured the extent of adaptation for two additional stimulus types that restrict the availability of specific cues. For these cue-restricted stimuli, which were randomly interleaved with cue-rich stimuli, correction trials were not used and no feedback was given. The first of these additional stimulus types comprised broadband noise with a random spectral profile that varied across trials (*Figure 1—figure supplement 1B*). These stimuli disrupt spectral localization cues because it is unclear whether specific spectral features are produced by the filtering effects of the head and ears or are instead properties of the sound itself (*Figure 1B*) (*Keating et al., 2013*). Consequently, if subjects adapt to asymmetric hearing loss by giving greater weight to the spectral cues provided by the non-deprived ear, we would expect to see less improvement in sound localization performance for random-spectrum sounds than for flat-spectrum sounds. This is precisely what we found (*Figure 1E,F*; random-spectrum slope values < flat-spectrum slope values; bootstrap test, p<0.01; Cohen's d = 1.18; see also *Figure 1—figure supplement 2*), indicating that adaptation involves learning to rely more on spectral cues.

However, if adaptation were solely dependent on this type of cue reweighting, we would expect no improvement in sound localization for narrowband sounds, such as pure tones. This is because spectral cues require a comparison of sound energy at different frequencies, which is not possible for these sounds (*Figure 1B*) (*Carlile et al., 2005*). Improved localization of pure tones would

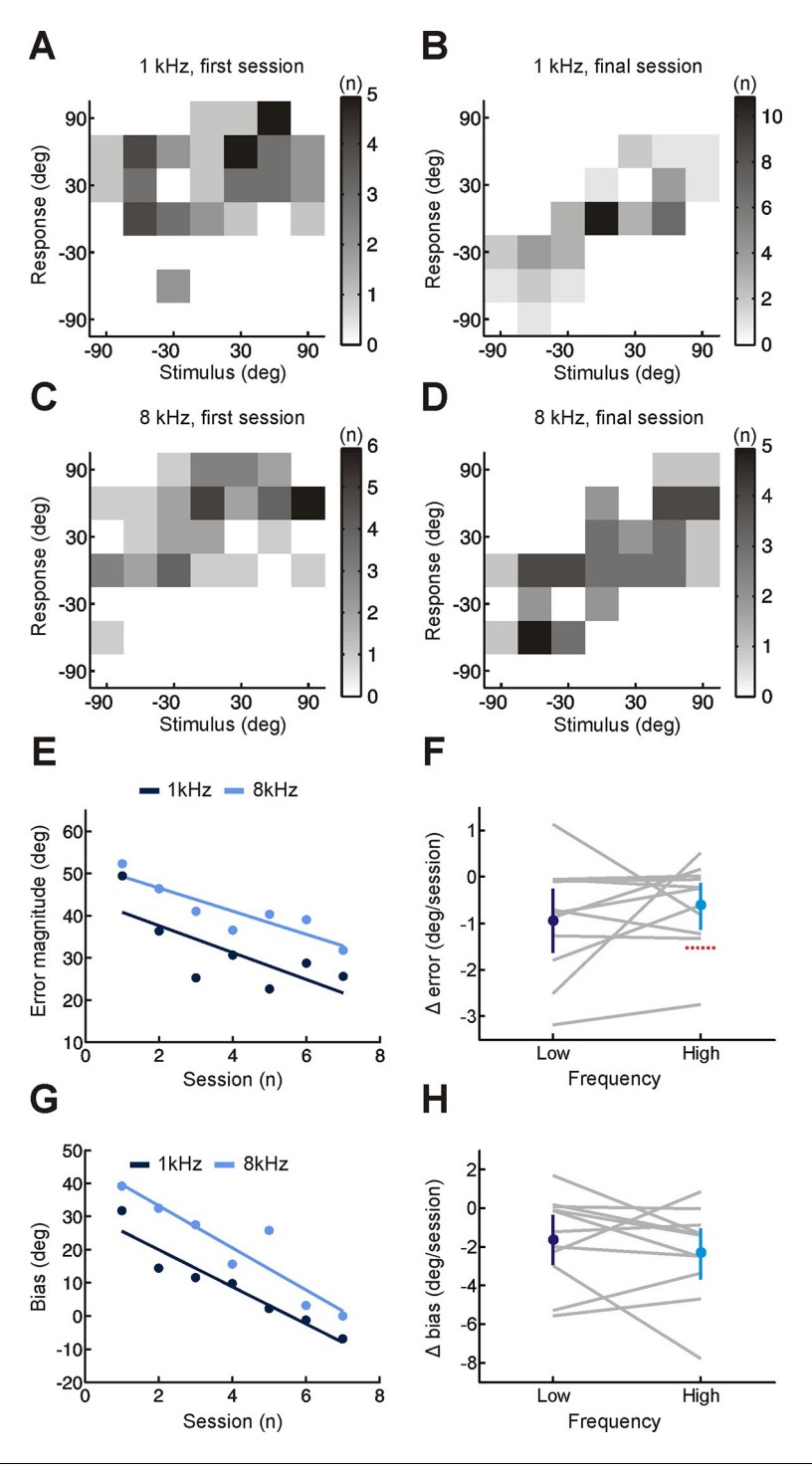

**Figure 2.** Effect of training on localization of pure tone stimuli in the horizontal plane by monaurally deprived human listeners. (A–D) Joint distributions of stimulus and response obtained from the first (A,C) and last (B,D) training sessions for low- (A,B) and high-frequency (C,D) tones. Data are shown for an individual subject wearing an earplug in the left ear, with grayscale indicating the number of trials corresponding to each stimulus-response combination. Because pure tones can be accurately localized only by using binaural spatial cues, which are susceptible to front-back errors, data from the front and rear hemifields have been collapsed. (E) Mean error magnitude plotted as a function of training session for the same subject shown in A–D. Data are plotted separately for low- (1 kHz, dark blue) and high-frequency (8 kHz, light blue) tones. Scores for each session (dots) were fitted using linear regression (lines) to calculate slope values, which quantified the change in error magnitude

*Figure 2 continued on next page*

*Figure 2 continued*

(Δ error) with training. Improved performance was associated with a reduction in error magnitude, producing negative values for Δ error. (**F**) Δ error for low- and high-frequency tones plotted for each subject (gray lines; n = 11). Mean values for Δ error across subjects (± bootstrapped 95% confidence intervals) are shown in blue. Although there are pronounced individual differences for the adaptation observed at the two tone frequencies, almost all values are <0, indicating that error magnitude declined over the training sessions. Dotted red line shows Δ error values that would have been observed if subjects had adapted as well as ferrets reared with a unilateral earplug (*Keating et al. 2015*; total Δ error reported for ferrets was divided by the number of training sessions used in the present study, n = 7; normalization used in previous work has been removed to facilitate comparison). (**G**) Bias in sound localization responses plotted as a function of training session for the subject in E. Positive values indicate that responses were biased toward the side of the open ear. Data are plotted separately for low- (1 kHz, dark blue) and high-frequency (8 kHz, light blue) tones. Scores for each session (dots) were fitted using linear regression (lines) to calculate slope values, which quantified the change in response bias (Δ bias) with training. Negative values of Δ bias indicate a shift in response bias toward the side of the plugged ear. (**H**) Δ bias for low- and high-frequency tones plotted for each subject (gray lines; n = 11). Mean values for Δ bias across subjects (± bootstrapped 95% confidence intervals) are shown in blue.

therefore indicate adaptive processing of binaural cues. Because interaural time differences (ITDs) and interaural level differences (ILDs) are respectively the primary cues for localizing low- (<1.5 kHz) and high-frequency (≥1.5 kHz) tones (*Blauert, 1997*), we tested each of these stimuli separately. To detect changes in binaural sensitivity, and facilitate comparison with previous work (*Keating et al., 2015*; *Kumpik et al., 2010*), stimulus and response locations in the front and rear hemifields were collapsed. This produces a measure of performance that is insensitive to front-back errors, which reflect failures in spectral, rather than binaural, processing. We observed improvements in subjects' ability to localize both low- and high-frequency pure tones over time, demonstrated by a decline in error magnitude (*Figure 2E,F*; Δ error <0; bootstrap test, p<0.01). The initial bias toward the side of the open ear was also reduced (*Figure 2G,H*; Δ bias <0; bootstrap test, p<0.01; low-frequency, Cohen's d = 0.7; high-frequency, Cohen's d = 0.96). Adaptation therefore involves a shift in the mapping of altered binaural cues onto spatial location. Together, these results show that subjects adapted to monaural deprivation using a combination of both cue remapping and cue reweighting.

We next considered the relationship between these two adaptive processes. Although cue remapping and cue reweighting share a similar time-course (significant correlation between the amount of remapping and reweighting across sessions; *Figure 3A*, r = 0.81, *P* = 0.028), the overall amount of cue remapping exhibited by each subject was independent of the amount of cue reweighting (*Figure 3B*, r = 0.03, *P* = 0.90). This inter-subject variability was not attributable to differences in the effectiveness of earplugs used (*Figure 3—figure supplement 1*). Instead, we found that these two adaptive processes are affected by the frequency composition of the stimulus in different ways (*Figure 3C*, interaction between sound frequency and adaptation type, p = 0.005, permutation test). As expected, cue reweighting was greater for frequencies where spectral cues are most prominent in humans (≥4 kHz, *Figure 3C*, p<0.05, post-hoc test; *Figure 3—figure supplement 2*) (*Blauert, 1997*; *Hofman and Van Opstal, 2002*), whereas equal amounts of cue remapping were observed for tones above and below 4 kHz (*Figure 3C*, p>0.05, post-hoc test).

This indicates that these adaptive processes are relatively independent of one another and suggests that they may depend on distinct neural substrates. This motivated us to revisit neurophysiological measures of cue reweighting and remapping in ferrets reared with an intermittent hearing loss in one ear (*Figure 3D*) (*Keating et al., 2013*; *Keating et al., 2015*). In common with our human behavioral data, we found no correlation between the degree of cue reweighting and remapping in cortical neurons recorded from ferrets raised with one ear plugged (*Figure 3E*, r = 0.08, p = 0.073). The type of plasticity observed also depended on the frequency preference of the neurons (*Figure 3F*, interaction between unit characteristic frequency and adaptation process, p = 0.012, permutation test). Greater cue reweighting was found in neurons tuned to frequencies where spectral cues are most prominent in ferrets (>8 kHz, *Figure 3F*, p<0.05, post-hoc test; frequency tuning bandwidth at 10 dB above threshold (μ ± SD) = 0.97 ± 0.51 octaves) (*Carlile and King, 1994*; *Keating et al., 2013*), whereas equal amounts of cue remapping occurred in neurons tuned to low

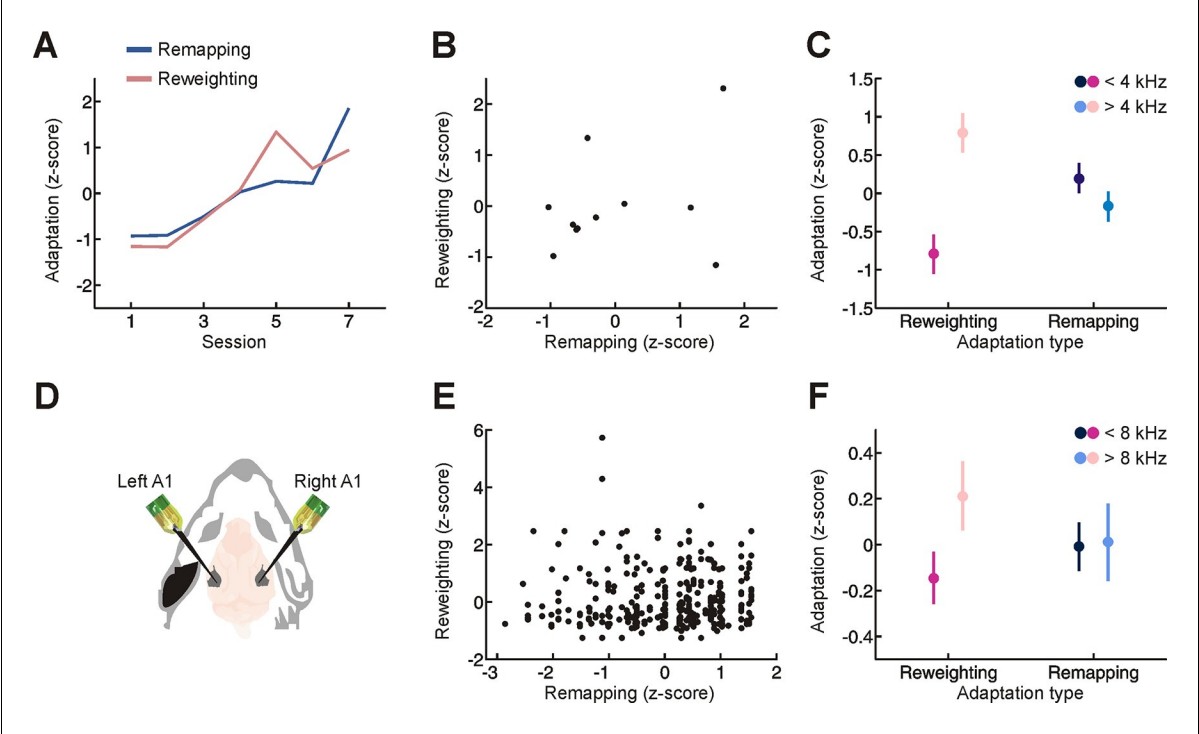

**Figure 3.** Relationship between different adaptive processes. (**A**) Time-course of behavioral adaptation for adult humans, measured by the amount of cue reweighting (pink) and remapping (blue). Data are normalized (z scores) to facilitate comparison between different adaptation measures. All data have been averaged across subjects. (**B**) Comparison between the amount of behavioral cue reweighting and remapping for individual human subjects (black dots; n = 11). Variation in the degree of adaptation across subjects was not attributable to differences in earplug effectiveness (*Figure 3—figure supplement 1*). (**C**) Amount of cue remapping (blue) and cue reweighting (pink) observed at frequencies above (lighter shades) and below (darker shades) 4 kHz. Greater reweighting of spectral cues (more positive values) is observed >4 kHz, which is where spectral cues are most prominent in humans. Frequency-specific measures of cue reweighting were determined using reverse correlation (see Materials and methods, *Figure 3—figure supplement 2*). (**D**) Bilateral extracellular recordings were performed in the primary auditory cortex of ferrets reared with an earplug in one ear. These data were then compared with controls to obtain measures of cue reweighting and cue remapping (see Materials and methods). (**E**) Cue reweighting versus cue remapping, with each dot representing either a single neuron or small multi-unit cluster (n = 505). (**F**) Amount of cue remapping (blue) and cue reweighting (pink) observed for neurons tuned to frequencies above (lighter shades) or below (darker shades) 8 kHz. To facilitate comparison between measures of cue reweighting and remapping at different frequencies, these values were normalized separately so that they each had an overall mean of 0 and a variance of 1. Greater reweighting of spectral cues (more positive values) is observed > 8 kHz, which is where spectral cues are most prominent in ferrets. Relative to humans, spectral cues in ferrets are shifted toward higher frequencies because of differences in head and external ear morphology.

The following figure supplements are available for figure 3:

**Figure supplement 1.** Variation across subjects in the degree of adaptation to acute asymmetric hearing loss is not related to differences in earplug effectiveness.

**Figure supplement 2.** Determining the behavioral importance of spectral features at different frequencies using reverse correlation.

and high frequencies (*Figure 3F*, p>0.05, post-hoc test). Thus, different neurons can exhibit cue remapping and reweighting in a relatively independent manner.

## Discussion

We have shown that adult humans can adapt to asymmetric hearing loss by both learning to rely more on the unchanged spectral localization cues available and by remapping the altered binaural cues onto appropriate spatial locations. Recent work has shown that both adaptive processes occur in response to monaural deprivation during development (*Keating et al., 2013*; *2015*). Our results suggest that this flexibility is likely to be a general feature of neural processing that also occurs in

adulthood. Moreover, we show that these two forms of adaptation emerge together and that remapping of binaural spatial cues occurs at low as well as high frequencies, indicating plasticity in the processing of both ITDs and ILDs.

Although adaptive changes in sound localization have previously been observed when human subjects wear an earplug for prolonged periods of everyday life (*Kumpik et al., 2010*), we found here that much shorter periods of training are sufficient to induce adaptation to an episodic hearing loss. Our results also demonstrate that subjects adapt using a combination of cue remapping and cue reweighting. In contrast, previous work has shown that cue remapping did not occur when subjects wore an earplug most of the time for several days, and were therefore able to interact with their natural environments under these hearing conditions, but received relatively little training (*Kumpik et al., 2010*). This suggests that the nature of adaptation may depend on the behavioral or environmental context in which it occurs. Consequently, it should be possible to devise training protocols that would help subjects to adapt to altered auditory inputs in ways that do not ordinarily occur, or occur more slowly, during the course of everyday life.

When both adaptive processes occur together, observed either behaviourally in adult humans or neurophysiologically in monaurally-deprived ferrets, there was no obvious relationship between the amount of cue remapping and reweighting. This is at least in part because the spatial cues involved differ in their frequency dependence. Whereas equal amounts of binaural cue remapping occurred at different frequencies, spanning the range where both ITDs and ILDs are available, reweighting of spectral cues was restricted to those frequencies where these cues are most prominent. This suggests that the neural substrates for cue remapping and reweighting are at least partially distinct, with separate populations of cortical neurons displaying different types of spatial plasticity depending on their frequency preferences and sensitivity to different spatial cues.

It is not known, however, whether remapping and reweighting occur at different stages of the processing hierarchy. Although experience-dependent plasticity in the processing of binaural cues has been observed at multiple levels of the auditory pathway (*Keating et al., 2015*; *Popescu and Polley, 2010*; *Seidl and Grothe, 2005*), the changes induced by unilateral hearing loss during development are more extensive in the cortex than in the midbrain (*Popescu and Polley, 2010*). Much less is known about the neural processing of spectral localization cues and how this might be affected by experience (*Carlile et al., 2005*; *Keating et al., 2013*). However, reweighting of these cues is likely to reflect a change in the way they are integrated with other cues, which is thought to occur in the inferior colliculus (*Chase and Young, 2005*). This is consistent with the finding that adaptive changes in sound localization behavior in monaurally deprived adult ferrets rely on descending projections from the cortex to the inferior colliculus (*Bajo et al., 2010*). It is likely therefore that adaptive plasticity emerges via dynamic interactions between different stages of processing (*Keating and King, 2015*).

Although we found evidence for both cue reweighting and cue remapping in our human behavioral and ferret neurophysiological data, the nature of the episodic hearing loss in each case was very different. Whereas ferrets had one ear occluded for ∼80% of the time over the course of several months of development (*Keating et al., 2013*; *2015*), adult human subjects wore an earplug for only ∼7 hr in total (1 hr every 1–3 days). It is not known whether comparable physiological changes to those observed in the ferrets are responsible for the rapid shifts in localization strategy in adult human listeners following these brief periods of acute hearing loss. Nevertheless, the close similarity in the results obtained in each species has important implications for the generality of our findings.

Our results emphasize the flexibility of neural systems when changes in sensory input affect ethologically important aspects of sensory processing, such as sound localization. They also reveal individual differences in the adaptive strategy adopted (*Figure 3B*). Further work is needed to understand the causes of these differences and to determine whether knowing how different individuals adapt to hearing loss could help tailor rehabilitation strategies. Our results also highlight the importance of training in promoting multiple adaptive processes, and this is likely to be relevant to other aspects of sensory processing (*Feldman and Brecht, 2005*; *Keating and King, 2015*; *Sengpiel, 2014*), particularly in situations where changes in sensory input affect some cues but not others.

## Materials and methods

All procedures involving human listeners conformed to ethical standards approved by the Central University Research Ethics Committee (CUREC) at the University of Oxford. All work involving animals was approved by the local ethical review committee and performed under licenses granted by the UK Home Office under the Animals (Scientific Procedures) Act of 1986. 11 audiologically normal human subjects (2 male, 9 female; aged 18–30) took part in the behavioral study. Sample size was determined on the basis of previous work, in which effect sizes of 2 – 4.6 were observed in human subjects who adapted to an earplug in one ear (*Kumpik et al., 2010*). To achieve a desired power of 0.8 with an alpha level of 0.001, 6–10 subjects were therefore required. All subjects provided written informed consent and were paid for their participation. Neurophysiological data were obtained from 13 ferrets (6 male, 7 female), seven of which were reared with an intermittent unilateral hearing loss, the details of which have been described previously (*Keating et al., 2013*). Briefly, earplugs were first introduced to the left ear of ferrets between postnatal day 25 and 29, shortly after the age of hearing onset in these animals. From then on, an earplug was worn ~80% of the time within any 15-day period, with normal hearing experienced otherwise. To achieve this, earplugs were monitored routinely and replaced or removed as necessary. All remaining ferrets were reared under normal hearing conditions. Expected effect sizes were less clear for neurophysiological changes so sample sizes were chosen based on previous studies in our lab (*Dahmen et al., 2010*).

For both human and animal subjects, hearing loss was induced by inserting an earplug into one ear (EAR Classic), which attenuated (low-pass filter, attenuation of 20–40 dB in humans, and 15–45 dB in ferrets) and delayed (150 µs in humans and 110 µs in ferrets) acoustical input (*Keating et al., 2013*; *Kumpik et al., 2010*). For 10 of the 11 human subjects tested, we measured hearing thresholds at 1–8 kHz in octave steps and assessed the impact on those thresholds of wearing an earplug in the trained ear (*Figure 3—figure supplement 1*). This yielded very similar results to those reported previously in humans (*Kumpik et al., 2010*).

### Human behavior

#### Apparatus

All human behavioral testing was performed in a double-walled sound attenuating chamber. Stimuli were presented to subjects using a circular array (1 m radius) of 12 loudspeakers (Audax TW025M0) placed at approximately head height, with loudspeakers positioned at 30° intervals (*Figure 1—figure supplement 1A*). This testing apparatus was similar to that used previously for both humans (*Kumpik et al., 2010*) and ferrets (*Keating et al., 2015*). Subjects sat at the mid-point of the loudspeaker array, with their head positioned on a chin-rest, and indicated the perceived location of each sound by using a mouse to click on a custom Matlab (Mathworks, Natick, MA) GUI that represented the locations of different loudspeakers. All stimuli were generated in Matlab, sent to a real-time processor (RP2; Tucker Davis Technologies), then amplified and routed to a particular loudspeaker using a power multiplexer (PM2R; Tucker Davis Technologies).

#### Stimuli

Stimuli consisted of either pure tones (varying in frequency from 1–8 kHz in one-octave steps) or broadband noise. All stimuli were 100 ms in duration (including 10 ms cosine ramps), generated with a sampling rate of 97.6 kHz, and presented at 49–77 dB SPL in increments of 7 dB. Different intensities and stimulus types were randomly interleaved across trials. Broadband noise stimuli (0.5–20 kHz) either had a flat spectral profile (flat-spectrum) or a spectral profile that varied randomly across trials (random-spectrum). Spectral randomization was produced by adding a vector to the logarithmic representation of the source spectrum (*Figure 1—figure supplement 1B*). This vector was created by low-pass filtering the spectra of Gaussian noise stimuli so that all energy was removed at frequencies > 3 cycles/octave (*Keating et al., 2013*). This removed abrupt spectral transitions to which humans are relatively insensitive (i.e. the width of any remaining peaks and notches exceeded 1/6[th] of an octave) (*Hofman and Van Opstal, 2002*). The RMS of this vector was then normalized to 10 dB.

These random-spectrum stimuli allowed us to determine which spectral features are behaviorally important (see *Figure 3—figure supplement 2*), whilst making very few assumptions about the nature of these features in advance. Their unpredictable nature also prevented subjects from

learning which spectral features were properties of the sound source. If subjects had learned that particular spectral features were invalid cues to sound location (i.e. they were not caused by the filtering effects of the head and ears and were instead properties of the sound source), they might have learned to ignore these features when judging sound location. This would have prevented us from measuring cue reweighting because our ability to do so requires subjects to misattribute spectral properties of the stimulus to the filtering effects of the head and ears.

## Training

Subjects were initially familiarized with the task under normal hearing conditions, receiving feedback for all stimuli. Once an asymptotic level of performance was reached, they were trained to localize sounds whilst wearing an earplug in either the left (8 subjects) or right (3 subjects) ear. Subjects completed 7 training sessions over ~3 weeks, with no more than 2 days between each session. Each session comprised ~800 trials and lasted ~45 min, with short breaks provided every ~15 min. Whilst undergoing training with an earplug in place, feedback was only provided for flat-spectrum broadband noise stimuli.

On trials where feedback was provided, correct responses were followed by a brief period during which the GUI background flashed green, with the GUI background flashing red for incorrect responses. The overall % correct score achieved for all feedback trials was also displayed by the GUI. Where feedback was given, incorrect responses were followed by "correction trials" on which the same stimulus was presented. Successive errors made on correction trials were followed by "easy trials", on which the stimulus was repeated continuously until subjects made a response. Recent work has shown that head-movements may enhance adaptation to changes in auditory spatial cues (*Carlile et al., 2014*). On easy trials, subjects were therefore allowed to move their heads freely until a response was made. Subjects were also not allowed to respond during the first 3 s of easy trials (i.e. any responses made during this period were ignored), which was visually indicated to subjects by the GUI background turning blue. In previous work, ferrets received broadly similar feedback when performing a sound localization task (i.e. incorrect trials were followed by correction trials and easy trials, with the latter allowing for the possibility of head-movements) (*Keating et al., 2013*; *2015*). However, instead of a GUI, ferrets received a small water reward for physically approaching the correct speaker location whilst the absence of water reward indicated an incorrect response.

## Analyses

Sound localization performance for pure tones was calculated by first collapsing stimulus and response locations in the front and rear hemifields. This was done to provide a measure of performance that is unaffected by front-back errors, which primarily reflect a failure in spectral, rather than binaural, processing. To facilitate comparison with previous work (*Keating et al., 2015*), the average error magnitude (mean unsigned error) was then used to quantify the precision of these sound localization responses. The mean signed error was also calculated to provide a measure of sound localization bias (*Kumpik et al., 2010*). Although we measured cue remapping at a number of frequencies above 1.5 kHz (2, 4 and 8 kHz), we found comparable training-induced changes in both bias (Kruskal-Wallis test, p = 0.18) and error magnitude (Kruskal-Wallis test, p = 0.58). These data were therefore pooled to facilitate comparison between cue remapping for tones above and below 1.5 kHz, which should respectively reflect changes in ILD and ITD processing (*Blauert, 1997*).

To assess the extent of cue reweighting at different frequencies, we used a method based on reverse correlation, which reveals the frequencies where spectral cues become more behaviorally important with training (*Figure 3—figure supplement 2*) (*Keating et al., 2013*). Note that the scale of the reverse correlation map (RCM) does not necessarily resemble that of the HRTF because the RCM is affected by the amount of spectral randomization present in stimuli (greater randomization typically produces larger RCM features) as well as the dependence on individual spectral features for localizing sounds in particular directions (i.e. if responses to a particular location can be induced by multiple spectral features or cues, then any given feature will not always be present when responses are made to that location; averaging over these data therefore reduces the scale of features detected by reverse correlation).

This analysis showed that training increased the behavioral importance of spectral cues at frequencies ≥4 kHz, but not below (*Figure 3—figure supplement 2*). In other words, we found greater

reweighting of spectral cues at higher frequencies. This is consistent with human head-related transfer functions, which show that spectral cues are most prominent at frequencies $\geq$4 kHz (*Blauert, 1997*; *Hofman and Van Opstal, 2002*). For frequencies above and below 4 kHz, we therefore calculated the average change in spectral feature strength separately, which provided a low- and high-frequency measure of cue reweighting. To facilitate comparison between different adaptive measures, we also separately calculated the average amount of cue remapping for tones above and below 4 kHz. Measures for different adaptive processes, which are expressed in different units, were then standardized by converting them to z scores.

## Neurophysiology

All neurophysiological procedures have been previously described in detail (*Keating et al., 2013*; *2015*). Bilateral extracellular recordings were made under medetomidine/ketamine anaesthesia from primary auditory cortex units (n = 505) in response to virtual acoustic space stimuli generated from acoustical measurements in each animal. These stimuli recreated the acoustical conditions associated with either normal hearing or an earplug in the left ear and were used to manipulate individual spatial cues independently of one another.

Cue weights were determined by calculating the mutual information between neuronal responses and individual spatial cues. A weighting index was then used to calculate the weight given by each neuron to spectral cues provided by the right ear (i.e. contralateral to the developmentally-occluded ear) relative to all other available cues. The mapping between binaural spatial cues and neurophysiological responses was measured by determining the best ILD for each unit, which represented the ILD corresponding to the peak of the binaural interaction function (see *Keating et al., 2015* for more details). Best ILDs and weighting index values were converted to z scores using the corresponding means and standard deviations of data obtained from controls. Data were normalized separately for each hemisphere and different frequency bands. Measures of cue reweighting and remapping for each unit therefore respectively reflected changes in weighting index values and best ILDs relative to those observed in controls. These values were then normalized again so that measures of reweighting and remapping had the same overall mean (0) and variance (1) prior to comparing the amount of each form of adaptation in different frequency bands.

Frequency tuning was calculated using 50-ms tones (0.5–32 kHz in 0.25 octave steps, varying between 30 – 80 dB SPL in increments of 10 dB). Characteristic frequency (CF) and bandwidth were calculated in a manner similar to that described previously (*Bartlett et al., 2011*; *Bizley et al., 2005*). Briefly, firing rates were averaged across stimulus repetitions (n = 30) of each combination of frequency and level. This matrix was then smoothed with a boxcar function 0.75 octaves wide, following which a threshold was applied that was equal to the spontaneous rate plus 20% of the maximum firing rate. CF was defined as the frequency that elicited the greatest response at threshold. Bandwidth was measured at 10 dB above threshold by first calculating the area underneath the tuning curve. We then identified a rectangle that had the same area but constrained its height to be equal to the maximum firing rate. The width of this rectangle then provided a measure of bandwidth that approximates the width at half-maximum for a Gaussian tuning curve (*Bartlett et al., 2011*).

## Statistical analyses

Confidence intervals at the 95% level were estimated empirically for different measures using 10,000 bootstrapped samples, each of which was obtained by re-sampling with replacement from the original data. These samples were then used to construct bootstrapped distributions of the desired measure, from which confidence intervals were derived. A bootstrap procedure was also used to assess the significance of group differences. First, the difference between two groups was measured using an appropriate statistic (e.g. difference in means, t-statistic, or rank-sum statistic). The data from different groups were then pooled and re-sampled with replacement to produce two new samples, and the difference between these samples was measured using the same statistic as before. This procedure was subsequently repeated 10,000 times, which provided an empirical estimate of the distribution that would be expected for the statistic of interest under the null hypothesis. This bootstrapped distribution was then used to derive a *P* value for the difference observed in the original sample. In all cases, two-sided tests of significance were used, with Bonferroni correction used to

correct for multiple comparisons. Cohen's d was also calculated to provide a measure of the effect size for different types of adaptation in adult humans.

The significance of factor interactions was also assessed using permutation tests (*Manly, 2007*). This involved randomly permuting observations across different factors and calculating an *F* statistic for each factor and interaction (i.e. the proportion of variance explained relative to the proportion of unexplained variance). This procedure was repeated many times in order to assess the percentage of repetitions that produce F values greater than those obtained for the non-permuted data. This percentage then provided an estimate of the *P* values associated with each effect under the null hypothesis. Precise details of the permutation procedure used have been described elsewhere (*Manly, 2007*). Additional comparisons between conditions were made using appropriate post-hoc tests corrected for multiple comparisons. Although bootstrap and permutation tests were used because they make fewer distributional assumptions about the data, conventional parametric and non-parametric statistical tests were also performed and produced very similar results (not reported).

## Acknowledgements

This work was supported by the Wellcome Trust through a Principal Research Fellowship (WT076508AIA, WT108369/Z/15/Z) to AJK.

## Additional information

### Competing interests

AJK: Reviewing editor, *eLife.* The other authors declare that no competing interests exist.

### Funding

| Funder | Grant reference number | Author |
| --- | --- | --- |
| Wellcome Trust | WT076508AIA | Andrew J King |
| Wellcome Trust | WT108369/Z/15/Z | Andrew J King |

The funders had no role in study design, data collection and interpretation, or the decision to submit the work for publication.

### Author contributions

PK, Conception and design, Acquisition of data, Analysis and interpretation of data, Drafting or revising the article; OR-P, Acquisition of data, Analysis and interpretation of data, Drafting or revising the article; JCD, OB, Acquisition of data, Drafting or revising the article; AJK, Conception and design, Drafting or revising the article

### Author ORCIDs

Peter Keating, http://orcid.org/0000-0002-0670-9075
Andrew J King, http://orcid.org/0000-0001-5180-7179

### Ethics

Human subjects: All procedures conformed to ethical standards approved by the Central University Research Ethics Committee (CUREC) at the University of Oxford. All human subjects provided informed written consent.
Animal experimentation: All procedures conformed to ethical standards approved by the Committee on Animal Care and Ethical Review at the University of Oxford. All work involving animals was performed under licenses granted by the UK Home Office under the Animals (Scientific Procedures) Act of 1986.

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
