## [Decision Letter]

Thank you for submitting your work entitled "Behavioral training promotes multiple adaptive processes following hearing loss" for consideration by *eLife*. Your article has been reviewed by three peer reviewers, and the evaluation has been overseen by Thomas Mrsic-Flogel as the Reviewing Editor and Gary Westbrook as the Senior Editor.

The following individuals involved in review of your submission have agreed to reveal their identity: Simon Carlile (peer reviewer).

The reviewers have discussed the reviews with one another and the Reviewing Editor has drafted this decision to help you prepare a revised submission.

Summary:

Your manuscript reports a largely human psychophysical study examining accommodation to transient monaural ear plugging and localisation training. The principal finding is that (i) cue reweighting to the intact open ear monaural spectral cues and (ii) cue remapping of the distorted ILD and ITD binaural cues to new spatial location both occur in this process of accommodation. Some previous electrophysiological data from ferret cortex is also included that concurs with the behavioural findings in humans. The reviewers agree that this is a well-crafted study that represents a logical step in a sequence of nice experiments from the King group at Oxford over the last few years. The experiments are well executed and the analysis is sound. The reviewers found the results not overly surprising, but they thought they were nonetheless important and worthy of wide dissemination.

Essential revisions:

The reviewers raised several important issues that we would like you to address in your revision.

1) The reviewers thought that the adult human psychophysics data were the most novel part of the manuscript. For this reason, they deserve to be show-cased more prominently, including expanding the analyses. For example, what is the advantage of inferring the degree of cue reweighting by presenting stimuli with random spectrum as opposed to stimuli where the spectrum is consistently adjusted to deviate from the flat spectrum by a known quantity? With random spectra, stimuli will occasionally provide a rich basis for determining azimuth location and other times not at all. The reviewers ask you to mine the dataset for performance differences in the random-spectrum condition for subsets of trials that offered rich vs. impoverished localization cues.

Alternatively, in order to make a much stronger statement about the contribution of cue reweighting versus remapping, experiments could be designed to bias the subjects towards one strategy or the other by training them with sounds that explicitly devalued monaural spectral cues or dichotic cues. If these data exist, please include them to the manuscript.

For comparison, it might be useful to compare the acute plugging human data to previous data on humans with chronic plugging. Please add if these data are available (additional experiments are not required).

2) There are significant and important differences in behavioural methods when comparing the human and ferret data. On the one hand, the manuscript draws attention to the fact that human accommodation training is based on a small number of relatively brief training episodes (in contrast to the chronic occlusion of previous studies). The demonstration of effective accommodation to this episodic exposure to the distorted cues is itself the most interest finding. On the other hand, the ferret electrophysiological data is obtained from animals reared with a chronic ear-plug since the onset of their hearing. Presumably, the training that these animals received also involved some form of feedback, however, those methods are not mentioned in the manuscript. More critical, however, are the differences in the chronic vs. transient nature of the exposure to the distorted cues. While it may be that both approaches are sufficient to induce the sort of cue reweighting and remapping that is argued for, these differences and resulting potential confounds do need to be grappled with more directly in the Discussion. An extension of the discussion about the differences between the ferret and human data, and how this impacts the main conclusions, is a crucial point that all reviewers felt needs far more attention.

---

## [Author Response]

*Essential revisions:*

*The reviewers raised several important issues that we would like you to address in your revision. 1) The reviewers thought that the adult human psychophysics data were the most novel part of the manuscript. For this reason, they deserve to be show-cased more prominently, including expanding the analyses. For example, what is the advantage of inferring the degree of cue reweighting by presenting stimuli with random spectrum as opposed to stimuli where the spectrum is consistently adjusted to deviate from the flat spectrum by a known quantity?*

We agree that it would be helpful to expand upon the advantages of random spectrum stimuli and describe their synthesis more clearly.

Firstly, our approach rests upon the fact that subjects find it difficult to determine whether monaural spectral features are properties of the sound source itself or whether they are caused by the acoustical filtering of the head and ears. It is this ambiguity that causes subjects to misinterpret spectral properties of the sound source as cues to sound location. And it is this misinterpretation that causes subjects to make systematic errors in sound localization. The extent and nature of these errors then allows us to determine how much weight subjects give to monaural spectral cues.

However, if we had used stimuli where the spectrum was consistently adjusted to deviate from a flat spectrum by a fixed quantity, subjects might have learned that this fixed deviation was a property of the sound source itself, rather than a result of the filtering effects of the head and ears. Consequently, subjects might have learned that a constant deviation from a flat spectrum was not a valid localization cue and ignored it when judging sound location. This would have prevented us from determining the weight given to spectral cues. In addition, adding a fixed spectral deviation to stimuli would have required us to make assumptions about the behavioural relevance of specific spectral features (i.e. we would need to know in advance which spectral deviations are likely to influence behaviour).

By varying the spectrum randomly across trials, and recording the spectrum used on each trial, we were able to use reverse correlation to empirically determine which spectral features are most relevant for sound localization. Spectral randomization also made it very difficult for subjects to learn which spectral features were properties of the sound source.

These advantages of spectral randomization are now stated clearly in the Methods:

“These random-spectrum stimuli allowed us to determine which spectral features are behaviorally important (see Figure 3—figure supplement 2), whilst making very few assumptions about the nature of these features in advance. […] This would have prevented us from measuring cue reweighting because our ability to do so requires subjects to misattribute spectral properties of the stimulus to the filtering effects of the head and ears.”

*With random spectra, stimuli will occasionally provide a rich basis for determining azimuth location and other times not at all. The reviewers ask you to mine the dataset for performance differences in the random-spectrum condition for subsets of trials that offered rich vs. impoverished localization cues.*

As noted by the reviewers, the usefulness of spatial cues can vary across trials when random spectrum stimuli are used. We attempted to minimize this trial-to-trial variability by constraining the overall amount of spectral randomization on each trial (SD = 10 dB). Nevertheless, within individual frequency bands, the amount of energy (sound level) and spectral randomization (SD of the amplitude values within a given frequency band) fluctuated across trials. This could have made some stimuli more difficult to localize than others by impoverishing certain spatial cues to varying degrees.

To test this, we separated our dataset into trials on which subjects responded correctly and trials on which they made errors. We then asked whether there were differences between the stimuli presented on correct versus incorrect trials. We found that the spectra of correctly localized stimuli were flatter than average (i.e. had lower spectral randomization), but only at high frequencies. We found no relationship between trial accuracy and variations of sound level within individual frequency bands. Together, these results confirm our claim that spectral randomization impairs sound localization primarily by limiting the usefulness of high-frequency spectral cues. This provides additional validation for our experimental approach and strengthens the case for using random-spectrum stimuli. We have therefore added this analysis to Figure 1—figure supplement 1.

*Alternatively, in order to make a much stronger statement about the contribution of cue reweighting versus remapping, experiments could be designed to bias the subjects towards one strategy or the other by training them with sounds that explicitly devalued monaural spectral cues or dichotic cues. If these data exist, please include them to the manuscript.*

We agree that this is an excellent idea and represents the next logical step in our attempts to understand these adaptive processes. In the present manuscript, we elected to focus on what happens when monaural spectral and dichotic cues are both useful, as is often the case in natural environments. In particular, we wanted to know what subjects would do spontaneously, without us providing feedback that biased them toward one adaptive strategy or the other. However, it would be interesting to determine what happens when specific cues are explicitly devalued during training. We do not currently have data of this type, and we believe it would require an entirely separate study to do justice to this approach. For example, there are a variety of different factors (e.g. frequency content) which could bias subjects towards using one strategy or the other, but it is unclear which of these is important. It is also possible that the importance of these different factors varies as a function of the task performed, or the environment in which it occurs. Addressing these issues properly represents a key challenge for future work but is beyond the scope of the current manuscript.

*For comparison, it might be useful to compare the acute plugging human data to previous data on humans with chronic plugging. Please add if these data are available (additional experiments are not required).*

We agree that it would be helpful to make direct comparisons with previous data, including the effects of chronic plugging in humans. We have therefore amended Figure 1 and Figure 2 to facilitate comparison with previous work in humans (Kumpik et al., 2010) and ferrets (Keating et al., 2015).

2) There are significant and important differences in behavioural methods when comparing the human and ferret data. On the one hand, the manuscript draws attention to the fact that human accommodation training is based on a small number of relatively brief training episodes (in contrast to the chronic occlusion of previous studies). The demonstration of effective accommodation to this episodic exposure to the distorted cues is itself the most interest finding. On the other hand, the ferret electrophysiological data is obtained from animals reared with a chronic ear-plug since the onset of their hearing. Presumably, the training that these animals received also involved some form of feedback, however, those methods are not mentioned in the manuscript. More critical, however, are the differences in the chronic vs. transient nature of the exposure to the distorted cues. While it may be that both approaches are sufficient to induce the sort of cue reweighting and remapping that is argued for, these differences and resulting potential confounds do need to be grappled with more directly in the Discussion. An extension of the discussion about the differences between the ferret and human data, and how this impacts the main conclusions, is a crucial point that all reviewers felt needs far more attention.

We agree that there are considerable differences between the ferret neurophysiology experiments and the human behavioural experiments, and that these differences should be discussed at greater length in the manuscript. Our intention was not to imply that the kinds of neurophysiological changes observed in ferrets necessarily underpin the behavioural changes we see in humans. Nevertheless, despite the methodological differences between our ferret neurophysiology and human behaviour, a surprising aspect of our data is the broad similarity of the findings from these experiments. All the same, whilst our neurophysiological data do provide a plausible neural substrate for our behavioural results, we primarily included the ferret neurophysiological data because they have important implications for the generality of our findings.

The methodological differences between the human and ferret studies are now spelled out more clearly in the Discussion:

“Although we found evidence for both cue reweighting and cue remapping in our human behavioral and ferret neurophysiological data, the nature of the episodic hearing loss in each case was very different. […] Nevertheless, the close similarity in the results obtained in each species has important implications for the generality of our findings.”

The feedback given to ferrets was conceptually very similar to that given to humans. In particular, incorrect trials were followed by ‘correction trials’ on which the same stimulus was presented. Incorrect responses to correction trials were also followed by ‘easy trials’, on which the stimulus was repeatedly presented until the ferret made its response. In addition, whilst ferrets were reared with a chronic earplug in one ear, they also experienced brief intermittent periods of normal hearing approximately 20% of the time. Both ferrets and humans therefore experienced episodic hearing loss, with the key difference being the temporal parameters that were used in each case.

These details are now included in the Methods:

“In previous work, ferrets received broadly similar feedback [to humans] when performing a sound localization task (i.e. incorrect trials were followed by correction trials and easy trials, with the latter allowing for the possibility of head-movements) (Keating et al., 2013; Keating et al., 2015). However, instead of a GUI, ferrets received a small water reward for physically approaching the correct speaker location whilst the absence of water reward indicated an incorrect response.”

“Neurophysiological data were obtained from 13 ferrets (6 male, 7 female), seven of which were reared with an intermittent unilateral hearing loss, the details of which have been described previously (Keating et al., 2013). Briefly, earplugs were first introduced to the left ear of ferrets between postnatal day 25 and 29, shortly after the age of hearing onset in these animals. From then on, an earplug was worn ~80% of the time within any 15-day period, with normal hearing experienced otherwise. To achieve this, earplugs were monitored routinely and replaced or removed as necessary.”